# SBO-RNN: Reformulating Recurrent Neural Networks via Stochastic Bilevel Optimization

**Ziming Zhang, Yun Yue, Guojun Wu, Yanhua Li, Haichong Zhang**

Worcester Polytechnic Institute
Worcester, MA 01609
{zzhang15, yyue, gwu, yli15, hzhang10}@wpi.edu

## Abstract

In this paper we consider the training stability of recurrent neural networks (RNNs), and propose a family of RNNs, namely SBO-RNN, that can be formulated using stochastic bilevel optimization (SBO). With the help of stochastic gradient descent (SGD), we manage to convert the SBO problem into an RNN where the feedforward and backpropagation solve the lower and upper-level optimization for learning hidden states and their hyperparameters, respectively. We prove that under mild conditions there is no vanishing or exploding gradient in training SBO-RNN. Empirically we demonstrate our approach with superior performance on several benchmark datasets, with fewer parameters, less training data, and much faster convergence. Code is available at `https://zhang-vislab.github.io`.

## 1 Introduction

**Training Stability in RNNs.** Recurrent neural networks (RNNs) have achieved significant success in learning complex patterns for sequential input data. Due to the repeatability of network weights in the chain rule when computing gradients, vanishing or exploding gradients often occur in training RNNs, *i.e.,* the gradient magnitudes either too small or too large, leading to the well-known critical training stability issue [Pascanu et al., 2013a]. To see this, suppose that we are optimizing the following RNN:

$$\min_{\omega,\theta} \mathbb{E}_{(x,y)\in\mathcal{X}\times\mathcal{Y}}\ell(h_T, y; \omega), \text{ s.t. } h_t = f(h_{t-1}, \mathbf{x}_t; \theta), \forall \mathbf{x}_t \in x, \forall t \in [T], \tag{1}$$

where $(x,y) \in \mathcal{X} \times \mathcal{Y}$ denotes the training data with samples $x = \{\mathbf{x}_1, \cdots, \mathbf{x}_T\} \subseteq \mathbb{R}^d$ and label $y$, $h_t \in \mathbb{R}^D$ denotes the hidden state variable at the $t$-th time step with a predefined initial $h_0$, $\ell, f$ denote the loss and nonconvex transition functions parametrized by $\omega, \theta$, respectively, and $\mathbb{E}$ denotes the expectation operation. The gradient of $\ell$ *w.r.t.* $\theta$ per data in the RNN can be computed as follows:

$$\frac{\partial \ell}{\partial \theta} = \frac{\partial \ell}{\partial h_T} \cdot \sum_{1 \leq t \leq T} \left( \frac{\partial h_T}{\partial h_t} \frac{\partial h_t}{\partial \theta} \right), \text{ where } \frac{\partial h_T}{\partial h_t} = \prod_{t < k \leq T} \frac{\partial h_k}{\partial h_{k-1}}. \tag{2}$$

The training challenge mainly comes from $\frac{\partial h_T}{\partial h_t}$, *i.e.,* $\|\frac{\partial h_T}{\partial h_t}\| \to 0$ leading to vanishing gradients and $\|\frac{\partial h_T}{\partial h_t}\| \to +\infty$ leading to exploding gradients, where $\|\cdot\|$ denotes the magnitude of a gradient.

**Penalty/Lagrangian Methods.** In fact, the learning problem in Eq. 1 is a constrained optimization problem with recursive equality constraints. To solve it, an intuitive idea is to relax it to an unconstrained optimization problem using, for instance, penalty methods [Zhang and Brand, 2017] or (augmented) Lagrangian methods [Gu et al., 2020] that lift the constraints into the objective function. As demonstration, we list a relaxation of Eq. 1 using a penalty method as follows:

$$\min_{\omega,\theta,\{h_t\}} \mathbb{E}_{(x,y)\in\mathcal{X}\times\mathcal{Y}} \left[ \ell(h_T, y; \omega) + \sum_{t\in[T]} \lambda_t \|h_t - f(h_{t-1}, \mathbf{x}_t; \theta)\|^2 \right], \tag{3}$$

35th Conference on Neural Information Processing Systems (NeurIPS 2021).

where $\lambda_t \geq 0, \forall t$ denotes a penalty coefficient. Now the gradient *w.r.t.* $\theta$ is only dependent on the second term, *i.e.,* least squares, with no training stability issue. Such relaxation has modified the RNN network architectures that leads to new challenges in such methods for RNN training, for instance:

- The size of variables $\{h_t\}$ increases linearly *w.r.t.* the sequence length $T$, leading to large memory.
- The learned parameters $\omega, \theta$ may not work well in the original RNN at test time, due to the penalty.

How to solve these challenges in the relaxed training approaches still remains an open question [Gu et al., 2020].

**Bilevel Optimization (BO) & Ordinary Differential Equations (ODEs).** Alternatively, based on a similar idea of penalty methods, a trivial yet equivalent modification of Eq. 1 is listed as Eq. 4 below, which essentially defines a bilevel optimization problem where one optimization problem is embedded within the other (*e.g.,* [Vicente and Calamai, 1994, Sinha et al., 2017]):

$$\min_{\omega, \theta} \mathbb{E}_{(x,y) \in \mathcal{X} \times \mathcal{Y}} \ell(h_T, y; \omega), \text{ s.t. } h_t = \arg \min_h \|h - f(h_{t-1}, \mathbf{x}_t; \theta)\|^2, \forall t \in [T] \tag{4}$$

$$\equiv \min_{\omega, \theta} \mathbb{E}_{(x,y) \in \mathcal{X} \times \mathcal{Y}} \ell(h_T, y; \omega), \text{ s.t. } \dot{h}_t = h_t - f(h_{t-1}, \mathbf{x}_t; \theta), \forall t \in [T], \tag{5}$$

where $\dot{h}_t$ denotes the change rate of variable $h_t$ over time in the *gradient flow*, and $\dot{h}_t = \mathbf{0}$ when $h_t$ converges to a fixed point, if exists. In fact, some of recent ODE based RNNs have similar transition functions to Eq. 5. For instance, Kag et al. [2020] proposed an incremental RNN (iRNN) with a transition function of $-\beta \dot{h}_t = \alpha (h_t + h_{t-1}) - f(h_t + h_{t-1}, \mathbf{x}_t; \theta), h_t(0) = \mathbf{0}$, and proved that there is no vanishing/exploding gradient in training iRNN. As we can see, the key difference between iRNN and Eq. 5 is that the current hidden state variable $h_t$ is introduced into function $f$, leading to a sequence of updates for $h_t$ through variable discretization. However, to the best of our knowledge, so far no work has been proposed to formulate RNNs using BO. Recently BO regains attention in meta-learning. For instance, Franceschi et al. [2018] proposed a framework based on bilevel programming to unify gradient-based hyperparameter optimization and meta-learning. Mounsaveng et al. [2021] proposed using online BO to tune the hyperparameters for data augmentation.

**Our Contributions.** *In contrast to the literature, in this paper we consider the training of RNNs itself as a bilevel optimization problem.* Specifically, we propose a new family of RNNs, namely *SBO-RNN*, that can be formulated using stochastic bilevel optimization (SBO), where we consider hidden state vectors as intermediate auxiliary variables that depend on the RNN model parameters. These auxiliary variables are optimized in the lower-level problem of SBO, and the model parameters as well as the predictor are learned by optimizing the outer objective function. We utilize stochastic gradient descent (SGD) and its momentum variants to solve such SBO problems, where in return the optimization process of the lower-level problem can be (approximately) represented by SBO-RNN. We also prove that under mild conditions, our SBO-RNN can avoid vanishing or exploding gradients in training by selecting proper learning rates for the lower-level problem. We conduct comprehensive experiments on several benchmark datasets to evaluate our approach, and achieve superior results.

In summary, our key contributions in this paper are listed as follows:

- We propose a new family of RNNs, namely SBO-RNN, that are the *first* in the literature, to the best of our knowledge, to formulate RNNs using stochastic bilevel optimization for streaming data.
- We prove that our networks manage to obtain good training stability by selecting proper learning rates for the lower-level problem to avoid vanishing and exploding gradients.
- We demonstrate superior performance empirically with fewer parameters, less training data, and much faster convergence.

## 2  Related Work

**RNN Architectures.** To address the training stability issue, there are significant amount of works on developing RNN architectures such as, just to name a few, long short-term memory (LSTM) [Hochreiter and Schmidhuber, 1997], gated recurrent unit (GRU) [Cho et al., 2014, Collins et al., 2016], unitary RNNs [Arjovsky et al., 2016, Jing et al., 2017, Zhang et al., 2018, Pennington et al., 2017], deep RNNs [Pascanu et al., 2013b, Zilly et al., 2017, Mujika et al., 2017, Zhang et al., 2021], linear RNNs [Bradbury et al., 2016, Lei et al., 2018, Balduzzi and Ghifary, 2016], residual/skip RNNs [Jaeger et al., 2007, Bengio et al., 2013, Chang et al., 2017, Campos et al., 2017, Kusupati

et al., 2018a], ODE RNNs [Talathi and Vartak, 2015, Niu et al., 2019, Chang et al., 2019, Kusupati et al., 2018a, Chen et al., 2018, Rubanova et al., 2019, Kag et al., 2020, Rusch and Mishra, 2021, Bai et al., 2019, Erichson et al., 2020], implicit RNN [Revay and Manchester, 2020], and TreeRNN [Lyu et al., 2021]. We summarize the literature as the following groups:

- *Recurrent units:* LSTM applies gate-controlled memory cells to mitigate the vanishing/exploding gradient issue in sequence-based tasks. GRU is another widely-used variant of RNNs, and similar to LSTM it can stabilize the training [Chung et al., 2014]. Recently, Jose et al. [2018] proposed a flexible Kronecker recurrent unit (KRU) based on Kronecker product. Nguyen et al. [2020] proposed MomentumRNN by integrating momentum into the recurrent unit.
- *Residual/Skip RNNs:* This family of RNNs feed-forward state vectors with the help of skip or residual connections to serve as a middle ground between feed-forward and recurrent models and to mitigate gradient decay. Such network architectures can also be used to search for equilibrium (or fixed) points in ODEs, such as iRNN [Kag et al., 2020].
- *Stable recurrent models:* Miller and Hardt [2019] proved that stable recurrent neural networks are well approximated by feed-forward networks for the purpose of both inference and training by gradient descent. The analysis is based on the contraction assumption in the state transition function. Revay and Manchester [2020] proposed an implicit model structure that allows for a convex parametrization of stable models using contraction analysis of non-linear systems. Collins et al. [2016] showed experimentally that all common RNN architectures achieve nearly the same per-task and per-unit capacity bounds with careful training, for a variety of tasks and stacking depths.
- *Unitary and orthogonal RNNs:* These RNNs aim to preserve the norm of hidden features (*i.e.,* $\|\frac{\partial h_t}{\partial h_k}\|$ in Eq. 2) by controlling the eigenvalues, explicitly or implicitly, that has been studied extensively in recent years [Arjovsky et al., 2016, Jing et al., 2017, Zhang et al., 2018, Pennington et al., 2017, Erichson et al., 2020, Kerg et al., 2019, Lezcano-Casado and Martınez-Rubio, 2019, Helfrich et al., 2018, Maduranga et al., 2019, Mhammedi et al., 2017]. For instance, Lezcano-Casado and Martınez-Rubio [2019] proposed expRNN by performing the first-order optimization with orthogonal and unitary constraints based on a parametrization stemming from Lie group theory through the exponential map.
- *RNNs with equilibrium hidden states:* In such RNNs, the hidden states are represented by the equilibrium points, if exist, of certain dynamical systems defined by ODEs, for instance. Typical works include AntisymmetricRNNs [Chang et al., 2019], FastRNN [Kusupati et al., 2018b], iRNN [Kag et al., 2020], Lipschitz RNN [Erichson et al., 2020]. It seems that all these works can prove good training stability from different perspectives such as generalization bound or norm of gradients, with empirical demonstration.
- *Deep RNNs:* RNNs are inherently deep in time. Inspired by this property, researchers are seeking to develop new networks to investigate the benefits of depth in space of RNN architectures. For instance, [Graves et al., 2013] combined multiple recurrent levels on the basis of bi-directional LSTM [Graves and Schmidhuber, 2005, Schuster and Paliwal, 1997] to improve RNN performance in speech recognition task. Another study in [Hermans and Schrauwen, 2013], with a deeper analysis of the different emergent time scales, also proposed a similar stacking architecture. Some deep RNNs have been proposed in the literature as well [Chen et al., 1995, El Hihi and Bengio, 1996, Fernández et al., 2007, Schmidhuber, 1992, Graves, 2013, Jaeger, 2007, Pascanu et al., 2013b, Pinheiro and Collobert, 2014, Li et al., 2018, Dennis et al., 2019].

**Optimization for RNNs.** Same as training other deep networks, SGD is widely used in training RNNs, but there is no guarantee of good training stability. In contrast, some works focus on optimization in RNNs to stabilize gradients. Typical works include truncated backpropagation through time (TBPTT) [Jaeger, 2002], real-time recurrent learning (RTRL) [Williams and Zipser, 1989], Frank-Wolfe algorithm [Yue et al., 2020], gradient clipping [Pascanu et al., 2013a, Li et al., 2018, Zhang et al., 2020], identity or orthogonal weight initialization [Le et al., 2015, Arjovsky et al., 2016, Jing et al., 2017, Jose et al., 2017, Mhammedi et al., 2017, Wisdom et al., 2016, Vorontsov et al., 2017], weight matrix reparametrization [Zhang et al., 2018]. Different from these works, we propose using SBO to interpret RNNs and further train them based on a two-loop algorithm where the inner loop solves the SBO problems and the outer loop optimizes the RNN parameters.

**Training Stability Analysis in RNNs.** The stability analysis of different network architectures often comes with the eigenvalues of the Jacobian of the hidden state dynamics, because RNNs can be considered as dynamical systems. For instance, Engelken et al. [2020] studied the Lyapunov spectra

---

**Algorithm 1** Training algorithm for SBO-RNN

---

**Input** : loss function $\ell$ parametrized by $\omega$, state transition function $F$ parametrized by $\theta$, step size $\eta$, training data $\mathcal{X}$ and labels $\mathcal{Y}$

**Output** : minimizer $\omega^*, \theta^*$

---

Randomly initialize $\omega^*, \theta^*$;
**repeat**
    /* lower-level optimization: RNN inference     */
    **foreach** $x \in \mathcal{X}$ **do**
        $h_0 \leftarrow \mathbf{0}$;
        **foreach** $t \in [T]$ **do**
            $h_t \leftarrow h_{t-1} - \eta \nabla F(h_{t-1}, \mathbf{x}_t; \theta^*)$;    // Or momentum SGD using Eq. 8 or 9
        **end**
    **end**
    /* upper-level optimization: hyperparameter learning     */
    $\omega^*, \theta^* \in \arg\min_{\omega, \theta} \mathbb{E}_{(x,y) \in \mathcal{X} \times \mathcal{Y}} \ell\left(h_T(\theta), y; \omega\right)|_{\omega = \omega^*, \theta = \theta^*}$;
**until** *Converge*;
**return** $\omega^*, \theta^*$;

---

of chaotic recurrent neural networks. Vogt et al. [2020] proposed using Lyapunov exponents to understand the information propagation in RNNs, but unfortunately there is no discussion on how to introduce such nice Lyapunov stability into the development of RNNs. Drgona et al. [2020] proposed viewing neural networks from a dynamical systems perspective as pointwise affine maps. However, the theoretical results are adapted from dynamical system analysis and the assumptions for deep neural networks are too strong to be met in practice. Tuor et al. [2020] proposed an ODE based network implementation to guarantee stability as well as incorporating prior knowledge. Recently, Ribeiro et al. [2020] proposed analyzing RNN training using attractors and smoothness as alternatives.

## 3 SBO-RNN

### 3.1 Stochastic Bilevel Optimization Formulation for RNNs

Recall that the goal of RNNs is to learn discriminant representations for different data sequences, regardless of how the intermediate hidden states are defined. In our approach, we first assume that a data sequence $x = \{\mathbf{x}_1, \cdots, \mathbf{x}_T\}$ is generated by a certain stochastic process with distribution $\mathcal{P}_y$ that is conditional on the label $y$ of $x$. That is, each $\mathbf{x}_t, \forall t \in [T]$ is sampled from $\mathcal{P}_y$, *i.e.,* $\mathbf{x}_t \sim \mathcal{P}_y$. We observe that this assumption can work well in practice, even for the sequences with strong dependencies among samples. Based on this assumption, we propose the following learning problem:

$$\min_{\omega, \theta} \mathbb{E}_{(x,y) \in \mathcal{X} \times \mathcal{Y}} \ell\left(h^*, y; \omega\right), \text{s.t. } h^* \in \arg\min_{h \in \mathcal{H}} \mathbb{E}_{\mathbf{x}_t \sim \mathcal{P}_y} F\left(h, \mathbf{x}_t; \theta\right), h_0 = \mathbf{0}, \forall \mathbf{x}_t \in x, \forall t \in [T], \quad (6)$$

where $F$ is a proper, differentiable everywhere, and lower-bounded function. As we can see, the lower-level optimization in Eq. 6 aims to learn the data representations $h$ on-the-fly, while the upper-level optimization aims to learn the hyperparameter $\theta$ of $F$ as well as the predictor $\omega$ simultaneously. Different from Eq. 4, for instance, our approach utilizes the stationary points of function $F$ as the representations of data sequences, and $\theta$ is learned so that such stationary points can be separated well for prediction. Similar ideas have been explored in the orthogonal RNNs such as expRNN [Lezcano-Casado and Martınez-Rubio, 2019] where the transition functions are nonexpansive *w.r.t.* hidden states (but no guarantee of convergence). It has been demonstrated that such nonexpansive constraints in learning can significantly improve the performance of RNNs. In this paper we further demonstrate that such convergence in the lower-level optimization can not only improve performance but also accelerate the training (see our experimental section).

### 3.2 Learning

**Lower-level Optimization.** We show our general training algorithm in Alg. 1, where the lower-level optimization is solved using SGD. We can also employ momentum SGD as our solver to train SBO-RNN. Specifically, below we list three popular variants of SGD, namely vanilla SGD, stochastic

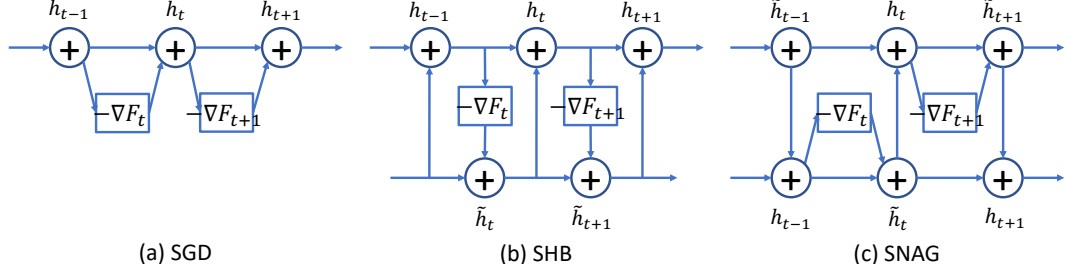

Figure 1: Illustration of SBO-RNN architectures using the optimizers in Eq. 7, 8 and 9, respectively.

heavy-ball (SHB) momentum, and stochastic Nesterov accelerated gradient (SNAG) momentum:

$$\text{SGD: } h_t = h_{t-1} - \eta \nabla F_t, h_0 = \mathbf{0}, h^* = h_T, \ \forall t \in [T], \tag{7}$$

$$\text{SHB: } \tilde{h}_t = \mu \tilde{h}_{t-1} - \eta \nabla F_t, h_t = h_{t-1} + \tilde{h}_t, \tilde{h}_0 = h_0 = \mathbf{0}, h^* = h_T, \ \forall t \in [T], \tag{8}$$

$$\text{SNAG: } \tilde{h}_t = h_{t-1} - \eta \nabla F_t, h_t = \tilde{h}_t + \mu(\tilde{h}_t - \tilde{h}_{t-1}), \tilde{h}_0 = h_0 = \mathbf{0}, h^* = h_T, \ \forall t \in [T], \tag{9}$$

where $\nabla$ denotes the gradient operator, $\nabla F_t = \nabla_h F(h_{t-1}, \mathbf{x}_t, \theta)$ denotes the current gradient *w.r.t.* $h$, $\eta \geq 0$ denotes the step size, and $\mu \in [0, 1]$ denotes the momentum parameter. Note that SHB and SNAG is identical to SGD when $\mu = 0$.

**RNN Interpretation.** Eq. 7, 8 and 9 define three different iterative methods that take the previous hidden state as input and output the current hidden state, leading to three different SBO-RNN network architectures (see Fig. 1). Note that our SBO-RNN shares similarities in some recent RNNs such as FastRNN [Kusupati et al., 2018b] and iRNN [Kag et al., 2020] that have skip connections in the architectures and have been demonstrated to be trained stably. Recently, momentumRNN [Nguyen et al., 2020] was proposed by drawing a connection between the dynamics of hidden states and GD. It is clear, however, that in terms of formulation the "momentum" here is only used for accumulating the weighted data samples without any purpose in optimization. In other words, its update rule has nothing to do with GD or momentum variants. In contrast, in our SBO-RNN we utilize first-order methods to really solve the lower-level optimization problems, which makes the gradient term in our SBO-RNN analytically integralable.

**Instantiations.** In this paper, we demonstrate our approach based on the following two functions:

$$F(h, \mathbf{x}_t; \theta) \overset{def}{=} \frac{1}{2} \left\| \alpha h - \phi \left( \mathbf{U}^T h + \mathbf{V}^T \mathbf{x}_t + \mathbf{b} \right) \right\|^2, \tag{10}$$

$$F(h, \mathbf{x}_t; \theta) \overset{def}{=} \frac{\alpha}{2} \|h\|^2 - \frac{1}{2} \left\| \mathbf{U}^{-1} \phi \left( \mathbf{U}^T h + \mathbf{V}^T \mathbf{x}_t + \mathbf{b} \right) \right\|^2, \tag{11}$$

where $\mathbf{U} \in \mathbb{R}^{D \times D}, \mathbf{V} \in \mathbb{R}^{d \times D}, \mathbf{b} \in \mathbb{R}^D, \alpha \in \mathbb{R}$ together form the model parameters $\theta$, $\phi : \mathbb{R}^D \to \mathbb{R}^D$ denotes an entry-wise nonconvex and differentiable function such as ReLU and $\tanh$, and $(\cdot)^T, (\cdot)^{-1}$ denote the matrix transpose and inverse operators. Here we borrow the transition function in vanilla RNNs that aim to enforce the hidden state vectors to satisfy certain properties induced by $\phi$. Note that in Eq. 11 we presume that $\mathbf{U}$ is full-rank and thus invertible. In practice we observe that this assumption holds all the time. Correspondingly, the gradients of both models are:

$$\nabla F_t = (\alpha \mathbf{I} - \nabla_h \phi_t)(\alpha h_{t-1} - \phi_t), \quad \nabla F_t = \alpha h_{t-1} - \phi_t, \tag{12}$$

where $\phi_t = \phi \left( \mathbf{U}^T h_{t-1} + \mathbf{V}^T \mathbf{x}_t + \mathbf{b} \right)$, and $\mathbf{I}$ denotes the identity matrix. Now we can substitute these gradients into Eq. 7, 8 and 9, respectively, to solve the lower-level optimization in Alg. 1. Note that Eq. 11 leads to a similar update rule for the hidden states to FastRNN [Kusupati et al., 2018b]. Without explicitly mentioning, in our experiments we will show the best performance from the two instantiations in Eq. 10 and 11.

**Convergence Rate.** Under some mild conditions, to minimize *strongly-convex and smooth* objectives, all the three stochastic optimizers can converge with the same rates as their deterministic counterparts [Assran and Rabbat, 2020]. In the *convex and smooth* setting, recent results in [Sebbouh et al., 2021] proved that their convergence rates can be arbitrarily close to $o(1/\sqrt{t})$, and can be exactly $o(1/t)$ in

the overparametrized case. In both *nonconvex and smooth* and *nonconvex and nonsmooth* settings, the convergence rates are changed to $O(1/\sqrt{t})$ [Yan et al., 2018, Mai and Johansson, 2020]. Overall, our solvers should be able to locate solutions around stationary points for our problem in Eq. 6.

**Network Implementation.** We illustrate three different network blocks for SBO-RNN in Fig. 1, where each directed edge towards $\bigoplus$ is associated with a certain learnable weight that is either 1 or related to one of $\alpha, \mu, \eta$ in Eq. 7, 8, 9 and 10. As we see, Fig. 1(a) is essentially a variant of ResNet [He et al., 2016], and Fig. 1(b) and (c) are similar to dual-path networks (DPN) [Chen et al., 2017] where information is propagated between two parallel paths. Note that though both (b) and (c) come from momentum SGD, the distributions of $h$ and $\tilde{h}$ are totally different due to their weighting strategies for gradients. The output $h_T$ is fed to a predictor parametrized by $\omega$ in Eq. 6.

Such networks can implement Alg. 1 effectively and efficiently. Precisely, we utilize the network blocks in Fig. 1 to construct a (sub)network to solve the lower-level optimization (*i.e.,* the foreach loop in Alg. 1), appended with a prediction subnetwork (*e.g.,* a fully-connected layer). The outputs of the entire network are used for minimizing the upper-level optimization problem. In training, we still use some popular deep learning solvers such as Adam [Kingma and Ba, 2014] to backpropagate gradients to update the network parameters, while the feedforward solves the lower-level optimization.

### 3.3 Analysis

**Theorem 1** (Upper Bound of Hidden Feature Distance during Network Inference). *Let $\forall t \in [T], x = \{\mathbf{x}_t\}, x' = \{\mathbf{x}'_t\}$ and $\{h_t\}, \{h'_t\}$ denote two data sequences and their corresponding hidden features, repectively. Suppose that function $F(h, \mathbf{x}_t; \theta)$ is differentiable, $L_h$-smooth w.r.t. $h$, i.e., $\|\nabla F(h, \mathbf{x}_t; \theta) - \nabla F(h', \mathbf{x}_t; \theta)\| \leq L_h \|h - h'\|$, and its gradient w.r.t. $\mathbf{x}_t$ is $L_x$-Lipschitz, i.e., $\|\nabla F(h, \mathbf{x}_t; \theta) - \nabla F(h, \mathbf{x}'_t; \theta)\| \leq L_x \|\mathbf{x}_t - \mathbf{x}'_t\|$. Then for SGD in Eq. 7, we have*

$$\|h_t - h'_t\| \leq \sum_{k \in [t]} \eta L_x (1 + \eta L_h)^{t-k} \|\mathbf{x}_k - \mathbf{x}'_k\| \leq t \eta \gamma_t L_x = O(t \gamma_t), \tag{13}$$

*where $\gamma_t = \max_{k \in [t]} \|\mathbf{x}_k - \mathbf{x}'_k\|$ and $\eta L_h \ll 1$.*

This result provides us a hypothesis on our faster convergence empirically that is beneficial from the factor $\gamma_t$ in the upper bound. Since we assume $\mathbf{x}_t \sim \mathcal{P}_y$, it is very likely that $\gamma_t$ for the data from the same class will be smaller than that from different classes, leading to clusters for different classes. We illustrate this intuition in Fig. 2, where red and green colors denote two different classes. The smaller upper bound encourages the data from the same class to form a cluster that can be easily separate from the data from different classes. This behavior would accelerate the learning of discriminative features in the networks, leading to fast convergence. Note that with simple algebra we can show that both $F$'s in Eq. 10 and 11 satisfy the smoothness conditions in Thm. 1.

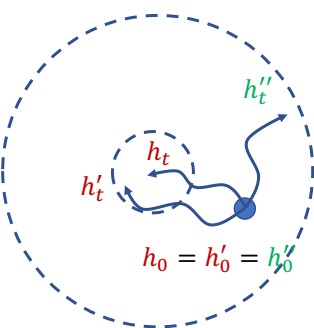

Figure 2: Illustration of clustering in the hidden feature space based on Thm. 1.

**Corollary 1.** *For SHB in Eq. 8 and SNAG in Eq. 9, supposing that the assumptions in Thm. 1 hold as well, then we still have $\|h_t - h'_t\| \leq O(t \gamma_t), \forall t \in [T]$.*

To see this, we can simply scale the upper bound with a factor of $T$.

**Theorem 2** (Training Stability). *Suppose that function $F$ is differentiable and $L$-smooth w.r.t. $h$, i.e., $\nabla^2 F \preceq L\mathbf{I}$ or $\|\nabla^2 F\| \leq L$ where $\mathbf{I}$ denotes the identity matrix. Then by properly choosing $\eta \in (0, \frac{\epsilon}{TL}]$ where $T$ is the sequence length and $\epsilon \in (0, 1)$ is a sufficiently small constant, we can guarantee that there is no vanishing/exploding gradient for training SBO-RNN using SGD in Eq. 7.*

**Corollary 2.** *For SHB in Eq. 8 and SNAG in Eq. 9, supposing that the assumptions in Thm. 2 hold, we still can guarantee that there is no vanishing and exploding gradient when training SBO-RNN.*

To see this, we can rewrite the update rules as accumulation of the previous gradients. Then based on the fact that $\frac{\partial h_{k-1}}{\partial h_{t-1}} = \mathbf{0}, \forall k \in [t-1]$ holds, we can still achieve $\frac{\partial h_t}{\partial h_{t-1}} \propto \mathbf{I} - \eta \nabla F_t$ that leads to the same conclusion in Thm. 2.

Table 2: Performance summary of different variants of our SBO-RNN on HAR-2. The training/test time is for per batch with 64 sequences.

| | dense SGD | dense SHB | dense SNAG | sparse SGD | sparse SHB | sparse SNAG |
|---|---|---|---|---|---|---|
| 1% data acc. (%) | **90.25± 0.71** | 86.59±0.74 | 87.19±0.14 | **79.18±1.45** | 78.55±1.45 | 77.89±1.41 |
| 10% data acc. (%) | 93.25± 0.07 | 93.44±0.67 | **93.52±0.38** | 90.76±0.32 | **91.98±0.49** | 91.89±0.59 |
| training time (s) | **0.60±0.07** | 0.65±0.06 | 0.67±0.10 | **0.54±0.02** | 0.55±0.04 | 0.58±0.04 |
| test time (s) | **0.30±0.02** | 0.33±0.05 | 0.31±0.02 | 0.28±0.01 | **0.26±0.01** | 0.27±0.01 |

# 4 Experiments

**Datasets.** We evaluate our method on long-sequence benchmarks with varying difficulties, and list the statistics of these datasets in Table 1. We strictly follow the experimental settings in the previous works (*e.g.,* [Kag et al., 2020, Kusupati et al., 2018b]) for fair comparison. Specifically, *Pixel-MNIST* refers to pixel-by-pixel sequences of images in MNIST

Table 1: Statistics and dimension of hidden state vectors in each benchmark dataset.

| Dataset | Statistics | | | | | Dim. |
|---|---|---|---|---|---|---|
| | #Train | #Test | #Time | #Feat. | #Cls. | $D$ |
| Pixel-MNIST | 60k | 10k | 784 | 1 | 10 | 128 |
| HAR-2 | 7.3k | 2.9k | 128 | 9 | 2 | 128 |
| PTB | 929k | 82k | 300 | 300 | 10k | 300 |

[LeCun and Cortes, 2010] where each $28 \times 28$ image is flattened into a 784 time-step sequence vector and normalized as zero mean and unit variance. *HAR-2* [Kusupati et al., 2018a] was collected from an accelerometer and gyroscope on a Samsung Galaxy S3 smartphone for human activity recognition tasks with zero mean and unit variance. *Penn Treebank (PTB)* dataset [Melis et al., 2017] consists of 300-word sequences for word-level language modeling task using the PTB corpus. The vocabulary consists of $10,000$ words and the size of trainable word embeddings is kept the same as the number of hidden units of the architecture.

**Baseline Algorithms.** We compare our results with baselines including vanilla RNN, LSTM [Hochreiter and Schmidhuber, 1997], AntisymmetricRNN [Chang et al., 2019], FastRNN [Kusupati et al., 2018b], FastGRNN [Kusupati et al., 2018b], ShaRNN [Dennis et al., 2019], IndRNN [Li et al., 2018], iRNN [Kag et al., 2020], LipschitzRNN [Erichson et al., 2020] and MomentumRNN [Nguyen et al., 2020]. Note that using the public code we have verified the results of each competitor on the datasets that were reported in the references. For simplicity and consistency we cite the numbers from the references, if exist, otherwise, we report our reproduced results with best tuned hyperparameters.

**Training & Testing Protocols.** In our implementation we utilize ReLU as our activation function (*i.e.,* $\phi$ in Eq. 10), as we observed that this activation works better than others such as $\tanh$ in terms of both accuracy and convergence. This observation is consistent with the state-of-the-art methods. We replicate the same benchmark training/testing split with $20\%$ of training data for validation to tune hyperparameters. Then we retrain the models using best tuned hyperparameters using the full training set and test them on the test set. We report our results (*i.e.,* accuracy and running time) over three trials with randomization wherever needed such as parameter initialization. All the experiments were run on an Nvidia GeForce RTX 2080 Ti GPU server.

**Hyperparameters.** We use the grid search to fine-tune the hyperparameters of each baseline as well as ours on the validation datasets whenever is necessary. We follow [Kusupati et al., 2018b, Kag et al., 2020, Chang et al., 2019] to set the hidden feature dimension $D$, as listed in Table 1. We further set $\eta = 10^{-3}$ in Eq. 7, 8 and 9 in all the experiments as we observe that this number works well and consistently that leads the lower-level optimization to converge. The batch size of 64 is used across all the datasets for all the methods. Adam [Kingma and Ba, 2014] is used as the optimizer for all the methods. The learning rate for training SBO-RNN architectures is always initialized to $10^{-3}$ with linear scheduling of weight decay.

## 4.1 Ablation Study on Adding Task and HAR-2 Dataset

The adding task is widely used for RNN evaluation. We strictly follow [Arjovsky et al., 2016] to generate the data using their public code. There are two sequences with length $T = 100, 500$, respectively. The first sequence is sampled uniformly at random $\mathcal{U}[0, 1]$. The second sequence is filled with 0 except for two entries of 1. The two entries of 1 are located uniformly at random

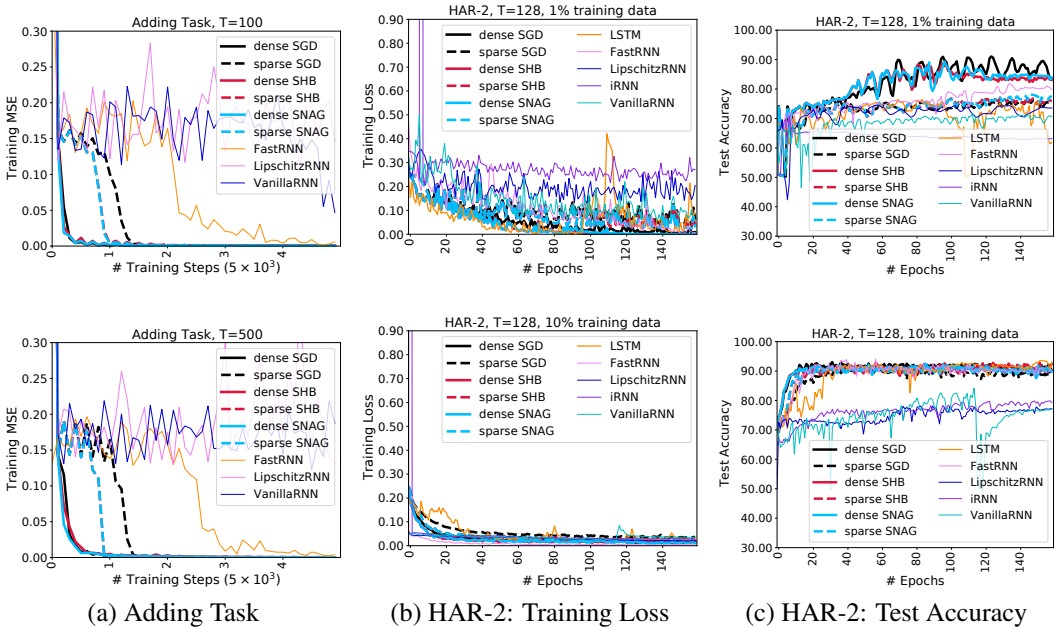

Figure 3: **(a)** Training loss on the Adding Task. **(b-c)** Training loss and test accuracy on HAR-2.

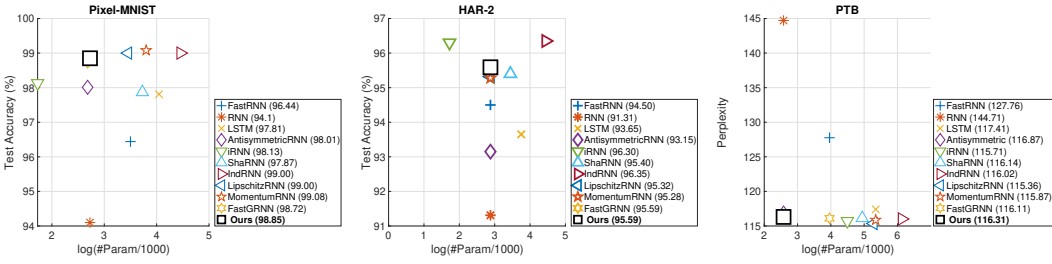

Figure 4: Benchmark comparison using different RNNs.

position $i_1, i_2$ in the first half and second half of the sequence. The prediction value is the sum of the first sequence between $[i_1, i_2]$. The ground-truth mean squared error is 0.167 when a model simply guesses 1 as the output regardless of the input sequence. We use the value 0.167 as the baseline.

We conduct comprehensive experiments based on $F$ defined in Eq. 10 to demonstrate the effectiveness of SBO-RNN, trained with mean-square-error (MSE) loss and hidden feature dimension $D = 128$, from the following three perspectives, compared with other baseline RNNs as summarized in Fig. 3 and Table 2:

- *Fewer parameters:* To this end, we simply set $\mathbf{U} = \beta \mathbf{I}$ where $\beta \in \mathbb{R}$ is a learnable parameter. In this way, we reduce the model size (including $\mathbf{U}, \mathbf{V}, \mathbf{b}, \alpha, \mu, \eta$) from $D(D \times d + 1) + 3$ to $D(d + 1) + 4$. We denote this variant "sparse" model in contrast to the default "dense" model. For instance, on HAR-2 our sparse models contain only 1.5K parameters, in contrast to the dense models with 17.9K parameters, that is more than one order of magnitude reduction. In terms of convergence, sparse models are slightly worse than the dense models, leading to worse test accuracy as well, generally speaking. We also try to implement a sparse variant of each competitor based on our setup, and compare these models on HAR-2. Using the whole training data, the test results for FastRNN, iRNN, LipschitzRNN, vanilla RNN and ours are 94.57%, 80.21%, 85.03%, 84.90%, and 94.96%, respectively, where our result is still the best. For the adding task, with the increase of $T$, the convergence behavior of SBO-RNN does not vary a lot in terms of number of training steps.
- *Less training data:* On HAR-2, using either 1% or 10% of the original training data, our method can outperform the competitors significantly by $\sim$10% in terms of test accuracy. Also in such

cases, our method converges faster when data is more, and our dense models seem to consistently work better than sparse ones. The number of data has more impact on sparse models than dense models, as the accuracy increases by $\sim$13% and $\sim$3%, respectively. Also, we observe that the hidden dimension has bigger impact on the performance for less training data. For instance, based on 1%, 10%, and full training data, we can achieve the test accuracy of 83.99%, 91.31%, 95.08%, respectively, with $D = 128$, in contrast with 86.29%, 92.37%, 94.74% with $D = 256$.

- *Faster convergence:* On both Adding Task and HAR-2 dataset, we clearly observe the much faster convergence of all of our models than the competitors, in terms of both training loss and the test accuracy. We hypothesize that such behavior has strong correlation with the lower-level optimization that enforces data to cluster, as shown in Thm. 1.

From Table 2, we can see that the running time of sparse models in both training and testing is significantly shorter than dense models, with smaller standard deviation as well (*i.e.,* better stability). Using the same setting, FastRNN, iRNN, LipschitzRNN, and vanilla RNN have the running time of 0.37x, 0.80x, 0.73x, 0.27x of our dense model, respectively, because our computational complexity per time step is higher (but $F$ in Eq. 11 has similar complexity to FastRNN). Also, from the table 2 we can see that the three optimizers for the lower-level optimization often achieve similar performance with small margins, especially when the training data is sufficiently large. Momentum variants bring more computation in both feedforward and backpropagation in training networks and inference. Therefore, in practice SGD based SBO-RNN seems to be sufficient. In terms of memory usage, they have 0.44x, 0.63x, 0.66x, 0.44x of our dense model.

In summary, our dense models outperform all the other models on Adding Task and HAR-2 dataset, and our sparse models seem to demonstrate strong potential in hardware implementation under resource-limited circumstances, achieving similar performance to those with larger model sizes.

## 4.2 State-of-the-art Comparison on Pixel-MNIST, HAR-2, and PTB Datasets

We compare the best performance of our dense models with other proposed baseline RNNs in Fig. 4. The x-axis measures the number of parameters in each network (including the classifier) in log-scale. Overall, our approach can always achieve comparable accuracy to the state-of-the-art, and for complex datasets such as PTB our approach can produce superior performance with smaller model sizes. Although the model sizes of our sparse models are much smaller (*e.g.,* $\sim$10x smaller than FastGRNN [Kusupati et al., 2018b] that was designed as a tiny-size RNN), the test accuracy is notably lower than the state-of-the-art, and thus we do not show them in the figure. Recall that our $F$ function is so simple that its variant is used in vanilla RNNs, while all the methods that achieve better accuracy, *e.g.,* IndRNN [Li et al., 2018], have very complicated hidden state transition functions, even with larger model sizes. In summary, our SBO-RNN has great potential to achieve the state-of-the-art with simpler transition functions and smaller model sizes.

## 5 Conclusion

In this paper we propose a new family of RNNs, namely SBO-RNN, that can be formulated using stochastic bilevel optimization. Using SGD or its momentum variants we convert such a bilevel optimization problem into an RNN architecture, where the lower-level optimization is mapped to the sub-network for computing hidden features and the upper-level optimization defines the predictor based on the computed features. We prove that under mild conditions there is no vanishing/exploding gradient in the training of our SBO-RNN, and thus our training is easy and stabilized. Furthermore, we demonstrate the superior performance of SBO-RNN on several benchmark datasets with faster convergence, even based on fewer parameters and less training data.

Currently, we lack strong theorems to explain the fast convergence behavior of SBO-RNN, though we have the hypothesis of data clustering, due to the lower-level optimization, that may significantly contribute to this. Also, our current setting has difficulty in adapting to deep RNNs such as IndRNN. In our future work, we will explore more on these topics.

**Acknowledgement.** Ziming Zhang and Yun Yue were supported in part by NSF grant CCF-2006738, Guojun Wu and Yanhua Li were supported in part by NSF grants IIS-1942680 (CAREER), CNS1952085, CMMI-1831140, and DGE-2021871, and Haichong Zhang was supported in part by NIH DP5-OD028162.

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
