# OpenReview forum: "SBO-RNN: Reformulating Recurrent Neural Networks via Stochastic Bilevel Optimization"
_NeurIPS.cc/2021/Conference — NeurIPS 2021 Poster_

### Official Review · Reviewer_Vvok · 2021-07-07

**Rating:** 4
**Confidence:** 4

**Summary:**

The authors propose viewing RNNs as meta-learning algorithms, where the RNN parameters optimize the hidden state. By using carefully chosen functions they show that reasonable RNN cells come out, like a residual RNN. They experimentally show to perform on par or better than competing approaches in 3 toy tasks.

**Main Review:**

Pros
- Seemingly good results on toy tasks.

Cons
- The authors make the paper unnecessarily difficult to follow by trying to draw parallel to meta-learning and optimization algorithms. The fact that the proposed RNN cells are the application of an optimization algorithm on a carefully chosen function doesn't add anything to the discussion. You could probably choose the function carefully  enough to get an LSTM or any other kind of RNN in the for of a meta-RNN. It's clear the authors have carefully chosen the function (Equation 10) to get an affine transformation as it's derivative (Equation 11).
- Theorem 2 should just be an observation. Obviously, if you have a residual connection which dominates, you're gradient won't vanish/explode. However, this again becomes harder to see if you make claims about meta-learning using the SGD optimizer.
- The paper lacks proper experiments. HAR-2, MNIST and the adding task are just toy problems that should be used to illustrate elements of the model. They can't be the only supporting experimental evidence.

**Time Spent Reviewing:**

4

---

> ### Author Response · Authors · 2021-08-10
> **Thanks for your valuable comments!**
>
> We will carefully modify the paper based on your comments in the final version. Below are our responses to your questions:
>
> **Q1: Carefully chosen function in Eq. 10** As we clearly stated in the paper in L177, "Eq. 10 borrows the transition function in vanilla RNNs". That is, we do not carefully choose a function for the experiments. For instance, another working function is $F = \frac{\alpha}{2}\|h\|^2 + \frac{1}{2}\mathbf{U}^{-1}\|\phi(\mathbf{U}^T h + \mathbf{V}^T \mathbf{x}_t + \mathbf{b})\|^2$ where $\phi$ is ReLU and the transition matrix $\mathbf{U}$ is assumed to be invertible (in practice we observe that this assumption is reasonable as it seems to hold always in training). We could find more complicated functions for meta-RNNs that satisfy the conditions in both theorems. We will add such discussion in the paper.
>
> **Q2: Theorem 2 should just be an observation. Obviously, if you have a residual connection which dominates, you're gradient won't vanish/explode.** We do not understand why such a training-stability theorem for RNNs should be an observation. Why does this "observation" hold? What does a residual connection dominate? Why is it obvious? Is there any place that people have shown/proven that this is true and obvious? We will appreciate it if the review can share more information about this comment.
>
> **Q3: More experiments** We are currently working on the PennTreebank (PTB) dataset (Melis et al., 2017), a popular language modeling task with sequence-to-sequence outputs. Within such very limited time, right now our method can achieve 120.35 in terms of perplexity with little parameter tuning, which we believe that we can further improve significantly. We are planning to add this experiment to the final paper.

---

> > ### Author Response · Authors · 2021-08-24
> > **Update for PTB experiment**
> >
> > Currently, our method can achieve 117.45 (previously 120.35), while under the same setting vanilla RNN, LSTM, FastRNN, FastGRNN can achieve 144.71, 117.41, 127.76, 116.11, respectively. We are still working on improving our result.

---

### Official Review · Reviewer_2xM4 · 2021-07-07

**Rating:** 5
**Confidence:** 3

**Summary:**

This paper formulates the problem of RNN training as a bilevel optimization problem in which the outer optimization is over the RNN parameters and the inner optimization is over the RNN activations. The authors prove upper bounds on the hidden feature distance of similar inputs. Moreover, the authors prove bounds on the inner optimization rate to guarantee training stability. Finally, the authors empirically find that their RNN formulation requires fewer parameters, less training data and converges faster than competing RNN formulations on benchmark problems.

**Limitations And Societal Impact:**

The authors adequately state the limitations of the work in the conclusion. There also do not appear to be potential negative societal impacts specific to this work.

**Main Review:**

**Originality**
This paper appears similar to prior works that view RNN hidden states as equilibrium points of a dynamical system (as noted by the authors in section 2). In this paper, the dynamical system over hidden states is itself an optimization. However, this contribution does not seem particularly novel since in prior work like iRNN, the dynamics of hidden states can also be viewed as an optimization (in which equilibrium points represent local minima of the lower-level optimization problem).

**Quality**
The theoretical claims appear correct and provide some justification for using meta-RNN, although it is not clear whether the results are specific to meta-RNN or whether similar results could be shown for standard RNNs.

The authors compare meta-RNN to several baselines. Unfortunately, the experimental results do not convincingly show that meta-RNN requires fewer parameters and less training data than baselines. Firstly, in Figure 5, meta-RNN-dense appears to require a similar number of parameters as baselines. Meta-RNN-sparse requires fewer parameters, but this does not appear to be a fair comparison with other dense methods such as regular RNN. A fair comparison would compare meta-RNN-sparse with sparse versions of baselines.

Moreover, the claim that meta-RNN requires less training data than baselines is not well supported by Figure 4. It appears that meta-RNN performs better than baselines in general for both the 1% and 10% training data settings on HAR-2. There doesn't appear to be evidence that meta-RNN is better than baselines *particularly* when training data is limited. This is partly because training curves for the full HAR-2 training data sets are not presented.

Finally, the datasets appear relatively simple. Conducting experiments on more challenging datasets would significantly improve this work.

In summary, the experiments could be further improved as follows:
1) Compare with sparse versions of baselines
2) Show that meta-RNN performs relatively better than baselines in regimes with low training data (vs. high training data)
3) Conduct experiments on more challenging datasets.


**Clarity**
The framing of the method in this paper as a type of meta-learning is slightly confusing since meta-learning typically refers to settings where a shared set of meta-parameters are found over several related datasets or tasks. In this paper, the shared set of meta-parameters correspond to a single task, which makes it unclear initially how this setting corresponds to meta-learning.

In general, meta-learning is a specific case of bilevel optimization in which an upper-level optimization problem is conducted over *several* different lower-level optimization problems. I believe that meta-RNN can be categorized as a type of meta-learning-  however, this needs to be further clarified by the authors.

In addition, it would help for the authors to clarify how their work conceptually differs from other RNN formulations with equilibrium hidden states. In section 3.2, the authors imply that prior works do not really solve their lower-level optimization problems- it would be ideal if the authors could be provide evidence for this.

There are also a few minor improvements that could be made:
1) If the authors wish to retain the meta-learning framing, it would help to refer to hyperparameters as meta-parameters
2) $\epsilon$ is not defined in Theorem 2
3) The authors state in section 3.1 that "Recall that the goal of RNNs is to learn discriminant representations for different data sequences, regardless of how the intermediate hidden states are defined.". This does not necessarily seem to be true: certain tasks may require similar representations for different data sequences. In general, the goal of RNNs depends on the optimization objective.

**Significance**
Currently, this contribution appears to be relatively incremental relative to prior RNN formulations that also use a dynamic update process to define hidden states. However, if experimental results and the clarity of the paper can be improved, the contributions of meta-RNN using fewer parameters, training data and training time could leave a significant impact in the research community.

**Time Spent Reviewing:**

2

---

> ### Author Response · Authors · 2021-08-10
> **Thanks for your valuable comments!**
>
> We will carefully modify the paper based on your comments in the final version. Below are our responses to your questions:
>
> **Significant flaws of the reviewer in understanding the originality of our paper**
>
> (1) *equilibrium points of a dynamical system vs. critical points of an optimization problem:* The reviewer seems to claim that they are equivalent, and hence our work is not novel as some other works like iRNN have done this. We totally disagree. For instance, a dynamical system $f(x) = 0$ and a minimization problem $\min_x \|f(x)\|$ are very likely to have totally different solution spaces (e.g. f(x) = 0 has no solution). How are the two problems equivalent???
>
> (2) *as noted by the authors in section 2:* Where did we note that our method "appears similar to prior works that view RNN hidden states as equilibrium points of a dynamical system"??? Could you please point out exactly where it appears in the paper?
>
> (3) *iRNN vs meta-RNN:* iRNN is defined using ordinary differential equations (ODEs), lead to a traditional constrained optimization with equations. In contrast, we formulate our meta-RNN using bilevel optimization that can be more general than iRNN.
>
> **not clear whether the results are specific to meta-RNN or whether similar results could be shown for standard RNNs???** In our statement of both theorems, we clearly state that both theorems are for the RNNs based on "SGD in Eq. 7", i.e. our meta-RNN.
>
> **Sparse model comparison** We conduct this experiment on HAR-2 using the whole training data, and the test results for FastRNN, iRNN, LipschitzRNN, vanilla RNN and ours are 94.57%, 80.21%, 85.03%, 84.90%, and 94.96%, respectively. Note that FastRNN accuracy is close to ours, because that when $\|\nabla_{h}\phi_t\|\ll\alpha$ in Eq. 11 holds, where $\|\cdot\|$ denotes the $\ell_2$ norm of a vector, then the gradient can be approximately computed as $\nabla F_t \approx \alpha^2 h_{t-1} - \alpha\phi_t$, which is reduced to a similar formula of FastRNN. However, our result is still the best among the competitors.
>
> **Show that meta-RNN performs relatively better than baselines in regimes with low training data (vs. high training data)???** We do not quite understand this comment. Doesn't Fig. 4(c) show such results?  The training curves for the full HAR-2 training data sets will be added to the paper, where our curves are similar to the ones in Fig. 4(c) using 10% training data and the curves of others are significantly improved.
>
> **Conduct experiments on more challenging datasets** We are currently working on the PennTreebank (PTB) dataset (Melis et al., 2017), a popular language modeling task with sequence-to-sequence outputs. Within such very limited time, right now our method can achieve 120.35 in terms of perplexity with little parameter tuning, which we believe that we can further improve significantly. We are planning to add this experiment to the final paper.
>
> **Differences from other RNN formulations with equilibrium hidden states** First of all, there is no work in the RNN literature using bilevel optimization formulation, including the RNNs with equilibrium hidden states, and thus there is no lower-level optimization in such RNNs. In contrast, our meta-RNN is the first to introduce bilevel optimization in formulation RNNs. This difference is fundamental.
>
> **Improving the paper** We will follow your suggestions to modify the paper.

---

> > ### Comment · Reviewer_2xM4 · 2021-08-28
> > **Thank you for your detailed response!**
> >
> > **Significant flaws of the reviewer in understanding the originality of our paper**
> > (1) Indeed, as the authors note, optimization procedures are different from dynamical systems. However, optimization procedures can be viewed as special cases of a dynamical system, with the dynamics of the optimization procedure defining the dynamical system. In this view, critical points of the optimization problem correspond to equilibrium points of the dynamical system.
> >
> > (2) Prior work involving RNNs with equilibrium hidden states is noted in the first paragraph of section 2 (the authors note the existence of these works, not their similarity with meta-RNN).
> >
> > (3) It is not clear to me how meta-RNN generalizes methods like iRNN; indeed, it appears that meta-RNN restricts the inner loop dynamical system to be an optimization while iRNN allows for more general inner loop dynamical systems.
> >
> > Nevertheless, the authors convincingly argue that this work is sufficiently novel compared to prior work since in meta-RNN, the inner loop dynamical system is itself an optimization procedure.
> >
> > **not clear whether the results are specific to meta-RNN or whether similar results could be shown for standard RNNs???**
> >
> > Yes; I am wondering whether it is possible to show similar results for standard RNNs, or whether the proof techniques rely on properties specific to meta-RNNs.
> >
> > **Sparse model comparison**
> > Thank you for these experiments!
> >
> > **Show that meta-RNN performs relatively better than baselines in regimes with low training data (vs. high training data)???**
> > That would be great, thank you! I believe with this graph, the claim made by the authors will be well supported.
> >
> > **Conduct experiments on more challenging datasets**
> > Thank you for this experiment!
> >
> > **Differences from other RNN formulations with equilibrium hidden states**
> > Thank you for clarifying!
> >
> > The authors convincingly argue for the originality of meta-RNN and moreover add additional experiments demonstrating the value of meta-RNN. Thus, I have increased my rating. However, I agree with the other reviewers about limited experimental results; I am willing to increase my rating further as the PTB experimental results improve.

---

> > > ### Author Response · Authors · 2021-08-30
> > > **Thanks for your reply!**
> > >
> > > **(1) optimization vs. dynamical system:** The reviewer seems to argue with us that "optimization procedures can be viewed as special cases of a dynamical system". We agree with this, in general. However, there are two points that we should make:
> > >
> > > (a) *An optimization problem (not algorithm or procedure) is NOT a dynamical system.* From this perspective, our bilevel optimization formula is novel, and nobody uses it in the literature.
> > >
> > > (b) *A dynamical system is NOT necessarily an optimization procedure/algorithm.* In the reviewer's original review, one claim is "However, this contribution does not seem particularly novel since in prior work like iRNN, the dynamics of hidden states can also be viewed as an optimization (in which equilibrium points represent local minima of the lower-level optimization problem).". This is a contradiction, and we do not agree entirely. This is also the answer to the reviewer's another question: **(3) It is not clear to me how meta-RNN generalizes methods like iRNN**, because meta-RNN (an optimization procedure/algorithm) CANNOT generalize iRNN (a dynamical system).
> > >
> > > **(2) Prior work involving RNNs with equilibrium hidden states is noted in the first paragraph of section 2 (the authors note the existence of these works, not their similarity with meta-RNN).** We will discuss the similarities and differences.
> > >
> > > **I am wondering whether it is possible to show similar results for standard RNNs, or whether the proof techniques rely on properties specific to meta-RNNs.** No, the results cannot be used for standard RNNs. The assumptions have been clearly shown in the theorems. So far we do not have any idea to generalize the results to other RNNs, including standard RNN. It will be highly appreciated if the review can share some thoughts with us on it!
> > >
> > > **PTB experimental results:** Currently, our method can achieve 117.45 (previously 120.35), while under the same setting vanilla RNN, LSTM, FastRNN, FastGRNN can achieve 144.71, 117.41, 127.76, 116.11, respectively. We believe that our current result has demonstrated the ability of our method in complicated tasks. We are still working on improving our result.

---

### Official Review · Reviewer_8SmP · 2021-07-16

**Rating:** 5
**Confidence:** 5

**Summary:**

This paper proposes a formulation of RNNs which allow practitioners to train RNNs using bilevel optimization algorithms. This is a novel perspective of RNNs.

The authors claim that their formulation of RNNs as an optimization process and their approach of training RNNs by solving a bilevel optimization problem prevents the problems of gradient vanishing/exploding. The authors demonstrate their methods on small datasets such as Pixel MNIST, Permutation MNIST, and HAR-2 (a dataset from phone accelerometer/gyroscope for activity recognition).


**Ethical Concerns:**

None.

**Ethics Review Area:**

["I don’t know"]

**Limitations And Societal Impact:**

Interesting novel perspective. Little to no practical impact for the present.

**Main Review:**

The paper provides an interesting and novel perspective of RNNs. I will first summarize the method. Let $h_t$ be an RNN’s state at time step $t$, then the standard forward propagation of RNNs would set $h_t = f(h_{t-1}, \theta)$ where $f$ is the parameterized function of the RNNs (such as an LSTM). Now, instead of enforcing the equality $h_t = f(h_{t-1}, \theta)$, this paper treats $h_t$ as a *free* parameter that should minimize the difference $|| h_t - f(h_{t-1}, \theta) || $. Under this formulation, the forward pass of an RNN can be treated as an optimization problem. This paper considers this optimization problem to be an inner loop of a bilevel optimization problem. Now, the entire RNN is supposed to minimize an objective function – for instance, predicting the correct MNIST pixel at each step. This objective function can be treated as an outer loop of the RNN’s bilevel optimization problem. This resulting bilevel optimization problem removes the recursive dependency of each $h_t$ on $h_{<t}$, and hence effectively avoids the RNN’s gradients from vanishing/exploding.

I personally find this bilevel optimization perspective of RNNs fascinating, novel, and interesting. In addition to this interesting formulation, the authors also derive an efficient analytical formula to iteratively solve their bilevel optimization problem using gradient-based updates (Equation 10-11). The authors also demonstrate that their meta-RNNs can be trained effectively to perform well on some small RNN tasks, namely Pixel-MNIST, Permuted-MNIST, and HAR-2.

While I like the perspective of this paper very much, I believe the following discussions and analysis should be included, in order for meta-RNNs to become useful:

1. Experiments on larger, more “real” situations such as large language models, should be conducted. There is also the Long-range Arena challenge, which measures how well RNNs can learn on long sequences. The authors can consider trying meta-RNNs on that challenge too.

2. Bilevel optimization is slow and finicky. The authors should compare the training time of meta-RNNs with other baselines. Table 2 does report the training time of meta-RNNs, but I cannot find any comparison of meta-RNNs’s training time and other methods. The comparison should be in controlled environments, e.g. on the same devices.

3. RNNs are no longer the state-of-the-art in sequence modeling. That title belongs to the Transformer model now. Can meta-RNNs achieve the same performance with Transformer? What are the pros and cons in comparison?

Overall, despite the novel perspective that this paper describes, I wish the authors had added larger and more “real” experiments to attest to the strengths of their model and method. Therefore, I am not certain whether this paper should be accepted or rejected.


**Time Spent Reviewing:**

1

---

> ### Author Response · Authors · 2021-08-10
> **Thanks for your valuable comments!**
>
> We will carefully modify the paper based on your comments in the final version. Below are our responses to your questions:
>
> **Q1: Experiments on language datasets** We are currently working on the PennTreebank (PTB) dataset (Melis et al., 2017), a popular language modelling task with sequence-to-sequence outputs. Within such very limited time, right now our method can achieve 120.35 in terms of perplexity with little parameter tuning, which we believe that we can further improve significantly. We are planning to add this experiment to the final paper.
>
> **Q2: Bilevel optimization is slow and finicky???**
>
> (1) Do not be afraid of bilevel optimization! Our Alg. 1 in the paper is an approximate solver for our bilevel formulation, whose complexity is linear to the number of time steps, same as all the other RNN competitors. However, due to the matrix multiplication in the gradient as shown in Eq. 11, our complexity per time step is $O(D^2)$ where $D$ is the hidden dimension. This complexity is higher than the competitors and has a significant impact on the running time and memory usage. However, we can overcome this issue in the following two ways, at least: (i) Supposing $\|\nabla_{h}\phi_t\|\ll\alpha$ in Eq. 11 where $\|\cdot\|$ denotes the $\ell_2$ norm of a vector, then the gradient can be approximately computed as $\nabla F_t \approx \alpha^2 h_{t-1} - \alpha\phi_t$, which is reduced to a similar formula of FastRNN. (ii) Change $F$ to another function that has no matrix multiplication in the gradient. For instance, if we set function $F = \frac{\alpha}{2}\|h\|^2 + \frac{1}{2}\mathbf{U}^{-1}\|\phi(\mathbf{U}^T h + \mathbf{V}^T \mathbf{x}_t + \mathbf{b})\|^2$ where $\phi$ is ReLU and the transition matrix $\mathbf{U}$ is assumed to be invertible (in practice we observe that this assumption is reasonable as it seems to hold always in training), then the gradient will become $\nabla_h F = \alpha h + \phi(\mathbf{U}^T h + \mathbf{V}^T \mathbf{x}_t + \mathbf{b})$. This again leads to an update rule for hidden features using SGD, similar to FastRNN. We will add such discussions in the final paper.
>
> (2) Therefore, using the same setting, FastRNN, iRNN, LipschitzRNN, and vanilla RNN have the running time of 0.37x, 0.80x, 0.73x, 0.27x of our dense model, respectively. Again we can improve the running time by optimizing either our code or the function $F$, which we have not done yet in current paper.
>
> **Q3: RNNs vs. Transformer** Transformers are a family of much more complicated RNNs with much more parameters but better parallelization. The key mechenism in transformers are the attention models that weight the samples at different time steps. It is like saving data in the buffer and learning with it. Similar usage of buffer has been explored in the RNN literature, for instance, Shallow RNN (Dennis et. al. NeurIPS 2019). Besides, there are several key differences between our method and transformers, at least:
>
> (1) Our focus is on the analysis of the training stability of meta-RNN, both theoretically and empirically, while there is no theoretical analysis of the training stability for transformers, to the best of our knowledge.
>
> (2) Our function $F$ is general, and if we can define a transformer mathematically using equations, then we are likely to model the transformer using a meta-RNN. This indicates that our meta-RNN may achieve similar performance to transformers if $F$ is sufficiently complicated. The computational complexity needs to be under control.
>
> Overall, this comment aligns very well with our ongoing work on the adaptation of our work to the transformer regime for better understanding its training stability as well as convergence. More investigations are needed in our future work.

---

> > ### Author Response · Authors · 2021-08-24
> > **Update for PTB experiment**
> >
> > Currently, our method can achieve 117.45 (previously 120.35), while under the same setting vanilla RNN, LSTM, FastRNN, FastGRNN can achieve 144.71, 117.41, 127.76, 116.11, respectively. We are still working on improving our result.

---

### Official Review · Reviewer_Qm4o · 2021-07-16

**Rating:** 5
**Confidence:** 3

**Summary:**

The paper considers the problem of training recurrent neural networks (RNNs) using an online bilevel optimization (OBLO). This allows to consider the problem as a meta-learning-like problem with streaming data, and obtain a new family of RNNs, dubbed as meta-RNN. The OBLO formulation is derived leading to two optimization problems that are solved interleaved. The first and inner stage is an inference step for the network (i.e., a forward pass) computes the hidden state out of the current network. Three versions are proposed to solve it based on stochastic gradient methods: SGD, SHB, and SNAG. Then, the convergence of these methods for this inner optimization task is analyzed theoretically to show the training stability in this inner optimization task. The second optimization aims to update the meta-RNN weights and hyperparameters. Last, the method is compared to several other RNN methods on four tasks: adding task, sequential MNIST and its permuted version, and HAR-2 (time series for activity recognition).


**Ethical Concerns:**

Not to the best of my knowledge.

**Limitations And Societal Impact:**

One important limitation that hasn't been addressed or mentioned by the authors is the fact that sequences can have variable length. This is very common in natural language for example. Another limitations not addressed is solving tasks that require sequential outputs.

Other limitations and societal impact have been mentioned to the best of my understanding.

**Main Review:**

The idea of looking at meta-learning with OBLO as an RNN is novel to me. The idea seems interesting. The paper cites a long list of papers. While doing an extensive review of related work is welcomed, it is always nice to explain how the current work differentiates from each one of these. In some cases, one may find that some of these works may not need to be cited or their detailed summary relegated to the supplementary material while using the space for the own technical aspects of this work.

The submission seems mostly technically sound. There are a few details that are unclear or seem missing. I would appreciate if the authors can respond to my questions below on these and other matters. The theoretical part of the paper is welcomed, however it is common to refer to the gradients of the weight updates (2 stage) for training stability of RNNs. Moreover, the main paper does not discuss the details of solving the network parameters stage. Another welcomed aspect is the extensive comparison against other methods in the experimental section. On the other hand, the authors forgot to discuss how to apply the model after trained to new (e.g., test) sequences, like those in the test sets in the experimental section. Regarding your deep-RNN limitation, you may find some inspiration in [1].

The paper seems nicely written and well organized. In general, the work is simple to understand. However, there some clarity details missing in the text that could improve the readability and reproducibility of this work. It is slightly confusing to go from RNN to meta-learning and back to meta-RNN in the introduction. Also, its motivation could be improved as well (what problem are we solving?). Referring to the parameters of the network with $\theta$ as the hyperparameters (for meta-learning) was confusing. There are some details described above that are missing (e.g., applying trained model to new sequences, solving the other optimization task, theorem assumptions met).

The methods presented in this work are relevant to the NeurIPS community. However, the potential value of this work seems small in my opinion. The missing details and the several limitations of this work are behind my score. I would be happy to reconsider my score if the authors could address my concerns.

Questions for the authors:
1. In Eq. (1), the RNN learning problem is defined as a constrained optimization problem. However, the constrain depends on $x$, which is part of the input distribution and changes with each input. It is unclear to me why this can be done. I would appreciate if the authors include the derivation that shows how they arrive to (1) from training.
2. Could you comment whether the conditions of the Theorems 1 and 2 are met after updating the network weights for your choice of $F$? How can one make sure that the method remains stable over batches? It would be very valuable to discuss this in the main paper.
3. How well does an LSTM perform in the HAR-2 10% and 1% of training data problem? (of similar hidden size and with a similar number of parameters.
4. How does your meta-RNN deal with problems that need to output sequences instead of a single prediction?
5. Both in training and testing(inference) time, what is the computational complexity (runtime and memory) of your method? How does it compare to training an RNN/LSTM  with SGD?

Minor:
* Please, consider removing the legend from inside the plots in Figure 4 to having it one time outside the plots.
* Consider increasing font size in plots.

References

[1] Turek et al., Approximating Stacked and Bidirectional Recurrent Architectures with the Delayed Recurrent Neural Network, ICML 2020.

**Time Spent Reviewing:**

8

---

> ### Author Response · Authors · 2021-08-04
> **Thanks for your valuable comments!**
>
> We will carefully modify the paper based on your comments in the final version. Responses to your questions are as follows:
>
> **Gradients of the weight updates for training stability of RNNs:** Our Thm. 2 exactly aims to address this problem in training networks, while Thm. 1 is for computing the features given the learned networks.
>
> **Details of solving the network parameters stage:** We provide the details in both Network Architectures in Sec. 3.2, Training & Testing Protocols in Sec. 4, and Hyperparameters in Sec. 4.
>
> **Test stage of learned networks:** Given the learned networks, the test stage is the same as vanilla RNNs and others by feeding the input data sequences into them. We will add this to the paper.
>
> **The potential value of this work seems small:** We provide a novel view by drawing connections between RNN training and meta-learning, supported with strong theorems and experiments. We develop a bilevel optimization framework as a principle that can be used to design new RNN architectures as guidance with a good training stability guarantee. Therefore, we believe that our potential value to the field is significant.
>
> **Q1:** Eq. 1 is a recursive way of defining RNN in a general form with a training objective. This definition is widely used in the RNN literature. For instance, letting $f$ be $h_t = relu(U^T h_{t-1} + V^T x_t + b)$ with $h_0 = 0$, then given $x_t$ at different time steps, we can compute $h_T$ recursively for a sequence with length $T$. Then we can plug $h_T$ into the loss in the objective function for learning.
>
> **Q2:** YES, function $F$ satisfies the conditions in both theorems. Recall that in practice, the data sequences are often upper bounded.
>
> (1) Given a learned network, we only need to consider variable $h$ (note that the activation function $\phi$ such as ReLU is well bounded as well). If $h$ can be upper bounded, then all the conditions in both theorems can be satisfied. Based on Eq. 11 for the gradient w.r.t. $h$, we can easily see that the gradient $\nabla F_t$ is upper bounded. Then within finite updates using Eq. 7, $h$ is guaranteed to be upper bounded. Within infinite updates ($T\rightarrow +\infty$), $h$ will converge to a local stationary point that has to be upper bounded. Therefore, function $F$ satisfies the smoothness conditions in both theorems.
>
> (2) For training stability in Thm. 2, the summation or mean over the batches does not change the lower and upper bounds of the magnitude of gradients in our proof, and thus the theorem still holds for the batch cases.
>
> **Q3:** We conduct the LSTM comparison on HAR-2 simply using the PyTorch function self.lstm = nn.LSTM(feature_shape, hidden_size). Using 1% and 10% training data, it can achieve ~75% and ~90% accuracy, respectively, which are both less than our results. Moreover, the training curves are very fluctuated, and hardly to converge. Besides, the LSTM model contains ~29K parameters while our dense model has ~7K parameters. We will add such results and curve comparisons in the final paper.
>
> **Q4:** If the ground-truth label $y$ is a structural output such as sequences, our framework should be able to deal with such cases theoretically. More investigations are needed on this topic.
>
> **Q5:** The computational complexity of our method given $F$ in Eq. 10 is $O(D^2)$ while vanilla RNN is $O(D)$, due to the fact that matrix multiplication is involved in gradient computation in Eq. 11. This complexity is higher than the competitors and has a significant impact on the running time and memory usage. However, we can overcome this issue in the following two ways, at least: (i) Supposing $\|\nabla_{h}\phi_t\|\ll\alpha$ in Eq. 11 where $\|\cdot\|$ denotes the $\ell_2$ norm of a vector, then the gradient can be approximately computed as $\nabla F_t \approx \alpha^2 h_{t-1} - \alpha\phi_t$, which is reduced to a similar formula of FastRNN. (ii) Change $F$ to another function that has no matrix multiplication in the gradient. For instance, if we set function $F = \frac{\alpha}{2}\|h\|^2 + \frac{1}{2}\mathbf{U}^{-1}\|\phi(\mathbf{U}^T h + \mathbf{V}^T \mathbf{x}_t + \mathbf{b})\|^2$ where $\phi$ is ReLU and the transition matrix $\mathbf{U}$ is assumed to be invertible (in practice we observe that this assumption is reasonable as it seems to hold always in training), then the gradient will become $\nabla_h F = \alpha h + \phi(\mathbf{U}^T h + \mathbf{V}^T \mathbf{x}_t + \mathbf{b})$. This again leads to an update rule for hidden features using SGD, similar to FastRNN. We will add such discussions in the final paper.

---

> > ### Comment · Reviewer_Qm4o · 2021-08-19
> > **Quick clarifications**
> >
> > I thank the authors for taking the time to address my questions thoroughly. Your responses would benefit the manuscript reproducibility.
> >
> > First, I would like to clarify my previous comment about the network parameters stage. My intention was referring to the “upper-level optimization” as referred in Algorithm 1.
> >
> > **Q1 follow up**: How does the constrain in Eq (1) consider the expectation $\mathbf{E}$ (for stochastic optimization) over the domain $(x,y)$?
> >
> > **Q2 follow up**: I’m not sure that I understood your answer. In the training case, $F$ is going to be updated after the upper-level optimization (where $\mathbf{U}$ is updated), how is $F$ still guaranteed to be L-smooth w.r.t. $h$ (Thm 2)?
> >
> > I agree with the other reviewers that a task with a real dataset is missing.
> > I would like to hear from the authors the perplexity obtained for the meta-RNN compared to LSTMs and other methods with the PTB dataset. Note that LSTMs achieve ~78 (with regularization) [1].
> >
> >
> > [1] S. Bai et al. “An Empirical Evaluation of Generic Convolutional and Recurrent Networks for Sequence Modeling”, 2018

---

> > > ### Author Response · Authors · 2021-08-20
> > > **Thanks for your comments!**
> > >
> > > **Q1 follow up** The expectation is over the training data. In stochastic optimization, the constraint should be satisfied for every training sample. Therefore, using the network architectures shown in Fig. 2, we can construct an RNN and its feedforward procedure solves the lower-level optimization in our algorithm (the foreach loop). Now we can train the network using Adam as our deep learning solver with batch learning.
> > >
> > > **Q2 follow up** First of all, our theorems are for inference, not for training, i.e. the network parameters are learned and fixed. Second, you are right that F will be updated after upper-level optimization, but still we can set $L$ as the upper bound over all possible $U$'s. As long as $U$ is upper bounded (equivalently $h$ is upper bounded), $L$ can be upper bounded as a scalar. This has no impact on our theorems.
> > >
> > > **PTB results** We are still working on improving our result (120.35). Note that *in [1] the sequence length is only 80, while our sequence length is 300 (longer is harder)*. This significant difference leads to a large gap in the results. Using the same setting as ours, vanilla RNN, LSTM, FastRNN, FastGRNN can achieve 144.71, 117.41, 127.76, 116.11. We will update our result.

---

> > > > ### Author Response · Authors · 2021-08-24
> > > > **Update for PTB experiment**
> > > >
> > > > Currently, our method can achieve 117.45 (previously 120.35), while under the same setting vanilla RNN, LSTM, FastRNN, FastGRNN can achieve 144.71, 117.41, 127.76, 116.11, respectively. We are still working on improving our result.

---

### Decision · Program_Chairs · 2021-09-28

**Decision:**

Accept (Poster)

**Comment:**

Overall this paper is not well written and it is hard to understand why the main algorithm (Algorithm 1) makes sense. It is also not clear if the relationship to meta-learning is useful. The results are on toy problems only plus on PennTreebank added during the rebuttal with performance much worse than ones reported in the literature for LSTM’s (on this dataset). It is good that there is a lot of relationship to literature, but the authors should provide in clearer derivation and intuitive explanation of the algorithm and resolved the discrepancy between their performance and the ones in literature.

Mini review from AC:
Even after the discussions I still don’t understand why Algorithm 1 makes sense. Intuitively, consider a sequence such as text. Inputs (x_t’s) at different time represent different pieces of information that need to be aggregated and processed using the rnn. One part of the text talks about something and another talks about something else. Why is h_t a gradient step from h_{t-1} of a function of x_t? Since x_t have completely different meaning at different times, how is this optimising anything? Is it because of assumption at line 142 and why this then possibly works so badly on PTB? Another thing unclear in the algorithm: The innermost loop in Algorithm 1 is a deterministic computation that depends on x and theta*. How does the h_T in the upper level optimisation then depend on theta (without star)? Or is it meant to be theta* and the theta, w minimization is done using bptt? Can you please provide full self-contained details of this algorithm together with the intuitive explanation why it makes sense?


**Consistency Experiment:**

NeurIPS has a long history of experimentation. In 2014, NeurIPS ran an experiment in which 10% of submissions were reviewed by two independent committees to quantify the randomness in the review process. This year, we repeated a variant of this experiment to see how the quality of the review process has changed over time.  This paper was part of the experiment and was therefore assigned to two committees (consisting of reviewers, an Area Chair, and a Senior Area Chair) that reached independent decisions.  If both committees made the same recommendation, this recommendation was followed. If a single committee recommended acceptance, the paper was accepted (with the exception of a few cases in which the other committee identified what we considered a fatal flaw, e.g., an error in a key result).

This copy’s committee reached the following decision: **Reject**

The other committee assigned to the paper recommended **Accept (Poster)**.  You can find the other set of reviews, along with any follow up discussion with the authors here:
https://openreview.net/forum?id=r1pprsDm185